# UKAT: Uncertainty-aware Kernel Association Test

## Abstract

Modern data collection methods routinely provide uncertainty estimates alongside point measurements, yet standard statistical tests typically ignore this valuable information. We introduce UKAT (Uncertainty-aware Kernel Association Test), a general framework for testing associations between variables while explicitly incorporating measurement uncertainties. UKAT treats each observation as a distribution characterized by its mean and uncertainty, then applies the Hilbert-Schmidt Independence Criterion (HSIC) to compare these distributional representations in kernel Hilbert spaces. Through extensive simulations, we demonstrate that UKAT achieves substantially higher statistical power than traditional association tests while maintaining proper Type I error control. We also validate UKAT's versatility across diverse scientific domains in proof-of-principle applications, including detecting prompt-induced effects in large language model responses on self-reported confidence and identifying associations in physical measurements with error estimates.

## 1 Introduction

Measurement uncertainty is ubiquitous across scientific disciplines (Figure 1). In high-throughput biological experiments, abundance estimates for genes, proteins, or metabolites carry uncertainties that reflect both technical factors, such as sample quality and sequencing depth, as well as the inherent stochasticity of biological processes. In astronomical surveys, photometric measurements of stellar properties often include error bars encoding detector noise, atmospheric conditions, and source brightness. Climate projections present similar challenges, where ensemble models generate temperature estimates with uncertainties reflecting both structural model differences and parameter estimation errors. In each domain, uncertainty quantifies data reliability and carries scientific meaning about experimental conditions, yet conventional analyses routinely ignore this rich information.

The neglect of measurement uncertainty comes at a significant cost. Traditional statistical tests assume homoscedastic errors, treating precise and imprecise measurements equally. This reduces statistical power for detecting associations, as noisy observations can dilute true signals (Figure 1a). Beyond power considerations, uncertainty patterns themselves can reveal scientifically meaningful relationships (Figure 1c). For instance, intrinsic and extrinsic gene expression noise in *E. coli* arises from functionally different sources and capture distinct biological heterogeneity.

Incorporating uncertainty into statistical testing faces practical challenges. Uncertainty estimates are themselves imperfect, arriving in diverse formats from bootstrap confidence intervals to Bayesian credible regions to self-reported confidence scores (Figure 1b). Additionally, the most general scenario involves uncertainty in both predictor and response variables, requiring joint modeling approaches (Figure 1d).

We present UKAT (Uncertainty-aware Kernel Association Test), a framework that incorporates measurement uncertainty into association testing through kernel methods. UKAT offers two key contributions: (1) generalized association testing for data with additional uncertainty estimates, accommodating various uncertainty types; (2) improved statistical power over traditional methods while maintaining proper Type I error control in the mean- and variance-only testing scenarios. Through extensive simulations and applications, we demonstrate UKAT's broad utility for uncertainty-aware statistical inference.

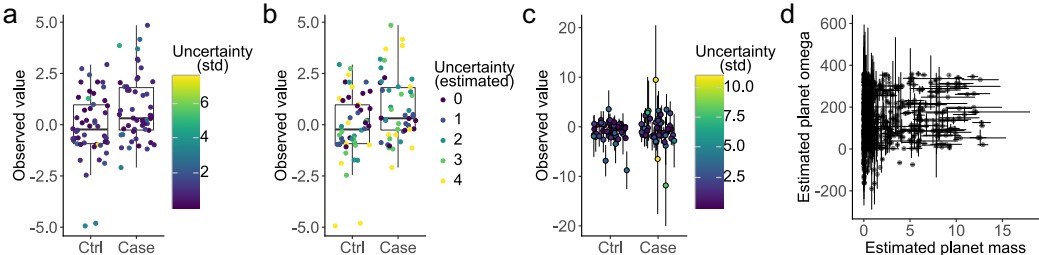

Figure 1: **Motivation for uncertainty-aware statistical testing.** (a) Per-observation measurement uncertainty can mask true group differences that become detectable when uncertainty is properly modeled. (b) Uncertainty estimates are imperfect and arrive in various formats, from bootstrap confidence intervals to self-reported confidence scores. (c) Some scientific questions focus on variance differences between groups, where incorporating uncertainty estimates can distinguish biological heterogeneity from measurement noise. (d) In general scenarios, both covariates and response variables carry uncertainty that requires joint modeling for optimal inference.

## 2 BACKGROUND

### 2.1 RELATED WORK

Several statistical approaches have attempted to address measurement uncertainty, though none provide a general framework for uncertainty-aware association testing.

Weighted statistical tests (Bland & Kerry, 1998; Pasek et al., 2025) represent one common approach, which typically assigns weights to observations inversely proportional to measurement variance (uncertainty). However, these weighting schemes are often arbitrary, and their effects on test validity remain unclear. When uncertainty manifests as probability distributions over discrete categories (e.g., genotype groups), Acar & Sun (2013) proposed a generalized Kruskal-Wallis test using probability-weighted rank sums with an asymptotic $\chi^2$ null distribution.

Multiple imputation (MI) (Little & Rubin, 2019) handles uncertainty by creating multiple plausible datasets and combining results across imputations. Originally developed for missing data, MI has been extended to genetic association studies with genotype imputation (Palmer & Pe'er, 2016) and differential expression testing based on model predictions with calibrated uncertainty (Sun et al., 2024). Briefly, a small number of imputed new datasets are independently drawn based on measurement uncertainty, which are then analyzed with standard statistical tests to characterize uncertainty-induced variation in the test results.

Random-effect meta-analysis (RMA) (Hedges & Vevea, 1998) combines effect estimates across studies while accounting for per-study standard errors. As the name implies, RMA models effect sizes as random effects whose variance components capture both within-study and between-study variation, effectively weighting studies by precision. To find association, mixed-effect models are typically used where study covariates are modeled in the fixed-effects term on the target effect size (Viechtbauer, 2010).

UKAT differs from these approaches in two major ways. First, we test associations involving *both* measurements and their uncertainties. When uncertainty doesn't vary with covariates, the problem reduces to robust mean (measurement) testing, for which we provide an alternative to weighted tests, MI, and RMA. Second, we don't assume uncertainty is perfect or follows any specific format, though when uncertainty represents standard deviation, our procedure has the intuitive interpretation of comparing distributions of *distributions*, as introduced in Figure 2.

### 2.2 KERNEL ASSOCIATION TESTS

Kernel methods detect nonlinear associations by embedding data into reproducing kernel Hilbert spaces (RKHS). The Hilbert-Schmidt Independence Criterion (HSIC) provides an elegant framework for testing independence between random variables $X$ and $Y$ by measuring the Hilbert-

Schmidt norm of the cross-covariance operator between their RKHS embeddings (Gretton et al., 2005a; 2007). The theoretical foundation rests on the fact that two random variables are statistically independent if and only if their covariance is zero (Gretton et al., 2005b),

$$\sup_{f \in \mathcal{F}, g \in \mathcal{G}} \mathrm{cov}(f(X), g(Y)) = 0$$

for all pairs of bounded continuous functions $(f, g)$. These functions can be studied in RKHS $\mathcal{F} = \mathrm{span}(\{k(\cdot, x), x \in \mathcal{X}\})$ associated with kernel $k(\cdot, \cdot)$, provided the RKHS is sufficiently rich (e.g., using *universal* kernels).

Formally, HSIC is the Hilbert-Schmidt norm of the cross-covariance operator mapping between RKHSs $\mathcal{F}$ and $\mathcal{G}$ associated with kernel functions $k_x$ and $k_y$ for variables $X$ and $Y$. With universal kernels, HSIC equals zero if and only if $X$ and $Y$ are statistically independent (Theorem 4 of Gretton et al. (2005a)). For empirical data with kernel matrices $K_X$ and $K_Y$, HSIC simplifies to

$$\mathrm{HSIC}(X, Y) = \frac{1}{(n-1)^2} \mathrm{tr}(K_X H K_Y H)$$

where $H = I - \frac{1}{n}\mathbf{1}\mathbf{1}^T$ is the centering matrix. Under the null hypothesis of independence, HSIC follows an asymptotic $\chi^2$ mixture distribution (Theorem 4 of Zhang et al. (2012))

$$T_{\mathrm{HSIC}} := \frac{1}{n}\mathrm{tr}(K_X H K_Y H) \xrightarrow[n \to \infty]{d} \sum_{i,j=1}^{n} \lambda_i \mu_j z_{ij}^2 \tag{1}$$

where $\{\lambda_i\}$ and $\{\mu_j\}$ are eigenvalues of centered kernels $H K_X H$ and $H K_Y H$, and $z_{ij}$ are independent standard Gaussian variables.

This framework provides the foundation for `UKAT`, which extends HSIC to handle uncertainty by constructing appropriate kernels that capture both measurement values and their associated uncertainties.

## 3 METHODS

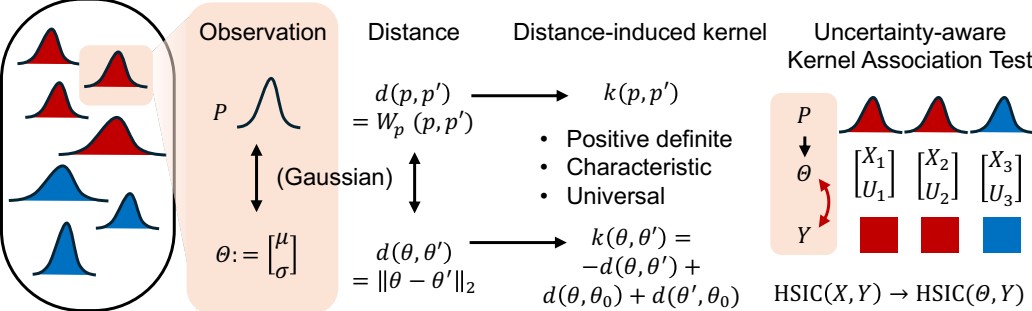

Figure 2: **UKAT conceptual framework.** `UKAT` considers each data point as a distribution characterized by both the measured value $X$ and its uncertainty $U$. Cross-sample pairwise distance is measured in the Euclidean space $\mathbb{R}^2$ using $\Theta := [X, U]$, which corresponds to the Wasserstein distance between Gaussian distributions. This distance induces an energy kernel suitable for HSIC-based association test.

### 3.1 FROM POINT ESTIMATES TO DISTRIBUTIONS

`UKAT`'s core innovation is treating each observation as a probability distribution rather than a point estimate (Figure 2). This conceptual shift enables kernel-based tests to detect associations between covariates and *distributions*, fundamentally departing from traditional tests that examine single properties like group means or variances. Under the null hypothesis, the distribution of distributions is independent of other covariates. This framework naturally encompasses conventional null hypotheses as special cases — for instance, equal group means (t-test) and equal group variances (F-test).

In real datasets, distributional information except for the observed value $X$ and its measurement uncertainty $U$, typically expressed as standard deviation or variance, is rarely available. For scalar variables, we denote $\Theta := [X, U] \in \mathbb{R}^2$ as the augmented observation. When only mean and variance are known, the Gaussian distribution maximizes entropy, making it a natural choice. We therefore characterize each observation as $N(x_i, u_i^2)$, assuming all relevant distributional information is captured in $\theta_i = [x_i, u_i]$.

A key insight is that for Gaussian observations, Euclidean distance in the augmented space $\Theta$ corresponds exactly to the Wasserstein distance between probability distributions

$$W_2(N(x_i, u_i^2), N(x_j, u_j^2)) = \sqrt{(x_i - x_j)^2 + (u_i - u_j)^2} = ||\theta_i - \theta_j||_2 := \rho(\theta_i, \theta_j). \quad (2)$$

This formulation naturally incorporates uncertainty into similarity computations. Observations with similar means but vastly different uncertainties are treated as dissimilar, reflecting that measurement precision affects statistical reliability. Conversely, observations with overlapping uncertainty ranges receive higher similarity scores, boosting variance-based inference.

## 3.2 Kernel Construction and Hypothesis Testing

Given the metric $\rho$ in (2), we construct a *distance-induced kernel*,

$$k(\theta, \theta') = -\rho(\theta, \theta') + \rho(\theta, \theta_0) + \rho(\theta', \theta_0) \quad (3)$$

which is induced by $\rho$ and centered at $\theta_0$. Setting $\theta_0 = 0$ yields the uncertainty-aware Wasserstein-based energy kernel

$$K_{ij} = -||\theta_i - \theta_j||_2 + ||\theta_i||_2 + ||\theta_j||_2. \quad (4)$$

Kernel choice determines the power of association tests and, specifically, which types of statistical dependency can be detected. Linear kernels capture correlations equivalent to multivariate correlation analysis, while nonlinear kernels like the energy kernel provide universal approximation properties, enabling detection of arbitrary dependencies given sufficient sample size.

**Proposition 3.1.** *The energy kernel in* (4) *is positive definite, characteristic, and universal.*

*Proof.* Universal kernels are stricter than characteristic kernels, which are both positive definite. We refer to Sriperumbudur et al. (2011) for detailed definitions. The positive definiteness follows from the fact that Euclidean space $(\mathbb{R}^d, || \cdot - \cdot ||_2^2)$ is of negative type (Proposition 3 and Lemma 12 of Sejdinovic et al. (2013)). Indeed, Euclidean spaces have strong negative type (Proposition 3.1 and Theorem 3.16 of Lyons (2013)), equivalent to the associated kernel being characteristic (Proposition 29 of Sejdinovic et al. (2013)). Universality follows from that $k$ is characteristic and translation invariant (Propositions 8 and 19 of Sriperumbudur et al. (2010)). $\square$

Alternative uncertainty-aware data kernels $K_X$ include RBF $K_{ij} = \exp(-\gamma d^2(\theta_i, \theta_j))$ and Laplacian $K_{ij} = \exp(-\gamma d(\theta_i, \theta_j))$, which are also universal but yield uncalibrated p-values and suboptimal power in our experiments.

For the covariate kernel $K_Y$, we use different approaches based on variable type. For categorical covariates, we employ the Dirac delta kernel

$$K_Y(i, j) = \mathbf{1}_{Y_i = Y_j}, \quad K_Y = \text{onehot}(Y)\text{onehot}(Y)^T$$

In two-sample scenarios, $Y_i \in \{0, 1\}$ indicates control and case groups. For continuous covariates, we use the linear kernel $K_Y = YY^T$.

**UKAT-C:** We refer to the test ((1)) with uncertainty kernel $K_X$ and standard covariate kernel as `UKAT-C`. P-values are computed using the Liu et al. (2009) moment matching approximation.

## 3.3 UKAT variants

**UKAT-R (Robust):** Uncertainty estimates from real data are often imperfect and arrive in various formats (Figure 1b). While noise in uncertainty $U$ does not directly interact with measurement $X$ in (2), arbitrarily large uncertainty scales can dilute signals and amplify estimation noise. To address

this, we replace the distance metric with a rank-based robust variant, yielding the Semblance kernel (Agarwal & Zhang, 2019),

$$K_{ij} = 1 - \frac{1}{2n}(d(\tilde{\theta}_i, \tilde{\theta}_j) + 2)$$

where $\tilde{\theta}_i = [\text{rank}(x_i), \text{rank}(u_i)]$ contains ranks across all $n$ observations, and $d(\cdot, \cdot)$ is Manhattan distance.

**UKAT-L (Linear):** Treating augmented observations $\Theta$ as regular 2D random variables, we can apply multivariate correlation analysis such as the RV coefficient (Robert & Escoufier, 1976) to study the linear association. This corresponds to a linear uncertainty-aware kernel,

$$K_X = XX^T + UU^T.$$

In two-sample scenarios with $m$ cases and $n - m$ controls, the test, `UKAT-L`, reduces to a multivariate extension of the two-sample t-test. Since observed values $X$ and uncertainties $U$ are isolated in separate dimensions, `UKAT-L` is equivalent to combining per-dimension t-test results under a new $\chi^2$ mixture null distribution.

**UKAT-G (General):** In the most general case, covariates also carry uncertainty (Figure 1d). Here, we apply uncertainty-aware energy kernels to both variables, testing associations between two distributions of distributions. While eigendecomposition of two full-rank kernels may be computationally expensive for large samples, this approach provides maximum expressiveness.

**Proposition 3.2.** *UKAT-G test statistic is zero if and only if $X$ and $Y$ are statistically independent.*

*Proof.* From Proposition 3.1, the uncertainty-aware energy kernel is universal. Applying Theorem 4 of Gretton et al. (2005a) completes the proof. □

## 4 RESULTS

### 4.1 SIMULATION STUDIES

We first evaluate `UKAT`'s performance on simulated two-group data following a parametric mixed-effect model, varying group means $(\mu_0, \mu_1)$ and uncertainty scales $(\sigma_0, \sigma_1)$. The mean and variance association tests examine null hypotheses $\mu_0 = \mu_1$ and $\sigma_0 = \sigma_1$, respectively, which are special cases of the distributional null $[\mu_0, \sigma_0] = [\mu_1, \sigma_1]$ detectable through `UKAT`.

For each observation, the noise-free mean $\mu$ and standard deviation $\sigma$ are drawn from

$$\sigma \sim \sigma_g \cdot \text{Exponential}(1), \quad \mu \sim N(\mu_g, 1),$$

and the observed value $x$ and estimated uncertainty $u$ follow

$$x \sim N(\mu, \sigma), \quad u \sim (1 - p) \cdot \sigma + p \cdot \text{Uniform}(u_{min}, u_{max}),$$

where $p$ denotes the noise level when uncertainty estimation is imperfect.

### 4.1.1 GROUP MEAN TESTING

We evaluated `UKAT`'s performance in detecting group mean differences, where uncertainty information should improve power by appropriately weighting observations. The simulation generated two groups of $n = 50$ observations each, with group means $\mu_0 = 0$ and $\mu_1 = 1$ but identical within-group uncertainty scales.

Figure 3 presents comprehensive simulation results comparing `UKAT` against traditional approaches across various uncertainty scenarios. Baseline methods include the standard t-test using only measured values $X$, its weighted variant with observation weights $w_i = \exp(-(u_i - u_{min})/(u_{max} - u_{min}))$, and a filtered version removing the top 20% of samples with highest uncertainty. We implemented a multiple imputation (MI) t-test following Sun et al. (2024), drawing 10 datasets from $x_i^{new} \sim N(x_i, u_i)$ and combining test results across imputations. A two-sample random-effects meta-analysis (RMA) was implemented using the Q-statistic from Hedges & Vevea (1998), which aligns with our data generation process and should achieve optimal power.

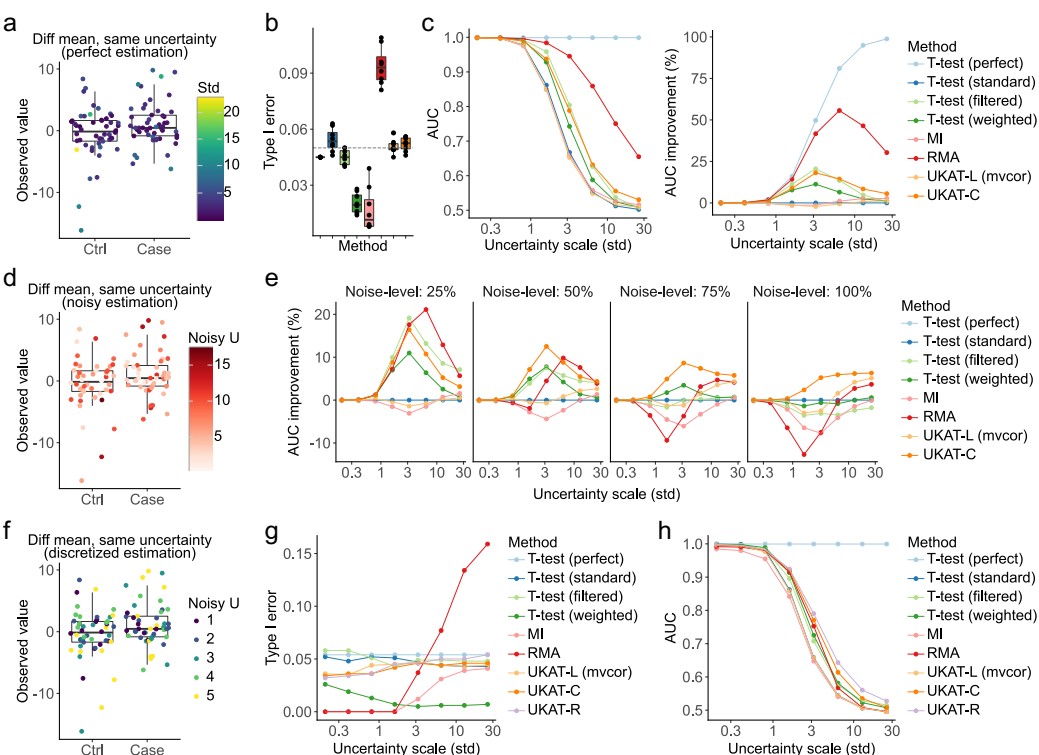

Figure 3: **UKAT improves power of group-mean tests by incorporating uncertainty.** (a) Simulated two-group data ($n_0 = n_1 = 50$) with different group means ($\mu_0 = 0, \mu_1 = 1$) but same group uncertainty scales. (b) Type I error control across methods at nominal $\alpha = 0.05$ level. (c) Area under ROC curve vs uncertainty scale, with AUC improvements computed over the standard T-test. T-test (perfect) represents ideal scenario with noise-free observations. (d-e) Performance under noisy uncertainty estimates with varying levels of estimation error. (f-h) Results when noisy uncertainty is discretized into categorical levels. MI: multiple imputation; RMA: random-effect meta analysis.

All proposed methods maintain proper Type I error control (Figure 3b) while incorporating uncertainty information. In contrast, RMA yields inflated Type I error consistent with literature reports (IntHout et al., 2014), while weighted and MI-based t-tests are overly conservative. As uncertainty scales increase, traditional t-tests lose power rapidly as noise overwhelms signal. When perfect uncertainty estimates ($u = \sigma$) are available, we ask whether each method can recover much of this power loss by explicitly modeling uncertainty structure. To account for false positive rate differences, we compared methods using area under the ROC curve, showing that RMA generates the highest power increase, followed by `UKAT` and the filtered t-test (Figure 3c). These results validate `UKAT` as a reliable alternative to weighted t-tests and data filtering, avoiding per-sample reliability adjustments that may compromise test validity. We confirmed that the asymptotic null distribution remains robust across sample sizes (Figure 7).

Real applications rarely provide perfect uncertainty estimates, motivating the noisy uncertainty simulations in Figure 3d. As expected, performance improvements from uncertainty modeling decay rapidly for most methods as noise levels increase (Figure 3e). The exception is `UKAT-C`, which remains robust and provides modest improvements even when uncertainty contains no additional information (noise level = 100%). Comparing the energy kernel of `UKAT-C` with the linear kernel of `UKAT-L`, this robustness likely stems from a regularization effect where large variations in $X$ are mitigated by random variations in $U$. Overall, our results suggest that proper modeling of noisy uncertainty can still provide detection benefit.

To further mimic real-world use cases, we designed a discrete uncertainty scenario (Figure 3f-h) where uncertainty can only be categorized into broad levels. Despite coarse discretization, `UKAT` maintains proper p-value calibration and retains significant power improvements over simple t-test

variants and more sophisticated MI and RMA baselines, suggesting broad utility even with qualitative uncertainty information. The translation-invariant kernel design makes `UKAT-C` robust to additive uncertainty estimation errors but not scaling errors. While this can be mitigated by rescaling $U$ to be comparable with $X$ (interpreting $U$ as the standard deviation of $X$), the rank-based `UKAT-R` eliminates scaling concerns and generates consistent, robust performance across noise levels (Figure 3h and Figure 8).

### 4.1.2 GROUP VARIANCE TESTING

We next explored `UKAT`'s performance for detecting variance differences, particularly relevant in applications like gene expression where differential variability indicates regulatory changes or cellular heterogeneity. The simulation maintained identical group means but varied uncertainty scales with ratio $r$. Incorporating per-sample uncertainty should improve power since it directly contributes to observed group-level variance.

Figure 4 shows that `UKAT` significantly outperforms F-tests in both false positive rate control and detection power. For baselines, we implemented a modified F-test where per-group variances were directly estimated from per-sample variances, which proved less powerful than testing uncertainty directly with t-test variants (including `UKAT-L`). Similar to the mean testing results, the noisy scenario (Figure 4d-e) confirms `UKAT`'s robustness even when uncertainty estimates contain substantial error. Comparing `UKAT-C` with `UKAT-L`, we again observe a likely regularization effect from measurement $X$ that mitigates large outliers in $U$.

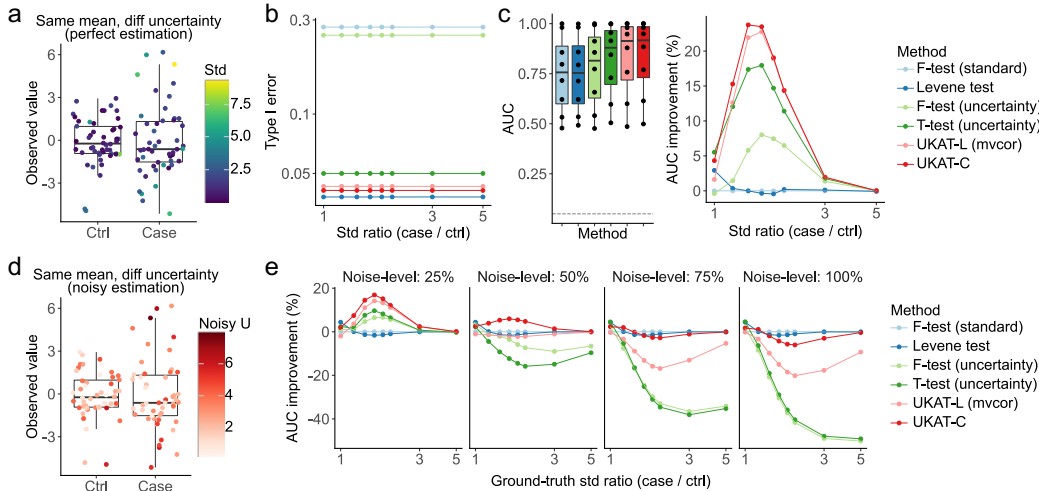

Figure 4: **UKAT improves power of group-variance tests by incorporating uncertainty.** (a) Simulated two-group data ($n_0 = n_1 = 50$) with same group means but different group uncertainty scales ($\sigma_0 = 1, \sigma_1 = r\sigma_0$). (b) Type I error control across methods. (c) Area under ROC curve vs uncertainty ratio $r$, with AUC improvements computed over the standard F-tests. (d-e) Performance under noisy uncertainty estimates demonstrates robustness to imperfect uncertainty quantification.

## 4.2 REAL-WORLD APPLICATIONS

We now demonstrate the importance of jointly testing both observed values and their measurement uncertainties in proof-of-principle applications on real data, showcasing findings that would otherwise remain hidden.

### 4.2.1 LARGE LANGUAGE MODEL (LLM) CONFIDENCE ANALYSIS

One potential solution to LLM "hallucinations" is encouraging models to honestly communicate uncertainty. We designed an experiment to test prompting effects on LLM response quality, specifically comparing standard, chain-of-thought (CoT, "think step by step to solve this problem"), and

double-check (DC, "calculate carefully, double-check your work") strategies. We evaluated GPT-5-nano performance on numerical reasoning tasks (Li et al., 2025). For each of 30 selected questions, we computed average accuracy based on 5 independent runs with identical prompts.

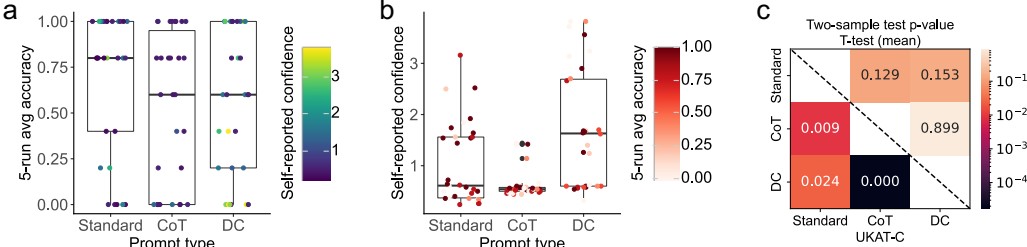

Figure 5: **UKAT reveals prompt effects on LLM confidence patterns.** (a-b) Three prompt types tested on GPT-5-nano for numerical reasoning tasks, measuring both 5-run accuracy and model self-reported confidence (0-10 scale). (c) Pairwise comparisons show that accuracy-only t-tests detect weak differences, while uncertainty-aware UKAT-C reveals highly significant effects driven by changes in self-reported model confidence. CoT: chain-of-thoughts; DC: double-check.

Figure 5 demonstrates how prompting strategies affect LLM behavior. Performance shows minimal effects from prompt strategies (Figure 5a). However, the double-check strategy significantly increases model-reported confidence, while chain-of-thought minimizes confidence variance (Figure 5b). These changes are readily detected by UKAT but missed by traditional accuracy-only tests (Figure 5c).

### 4.2.2 EXOPLANET PHYSICAL PROPERTIES

Figure 6 showcases another application to astronomical data[1], where both measured values and estimation errors are available for various exoplanet physical properties and orbital characteristics. We performed association tests for all pairs of properties and document below unexpected associations that are only significant when uncertainty is incorporated.

Figure 6a examines planet radius versus orbital eccentricity, which theoretically should show no association, consistent with the negligible Pearson correlation. However, UKAT detected that smaller planets tend to have higher eccentricity estimation errors, reflecting increased characterization difficulty that may introduce observational biases in exoplanet catalogs. Figure 6b presents a more complex scenario involving planet mass and argument of periastron. While traditional correlation detects weak association (Pearson $r = 0.10$, $p = 0.01$), the general association is much stronger, driven by negative relationships between mass and omega estimation uncertainty, again suggesting potential systematic biases in the data.

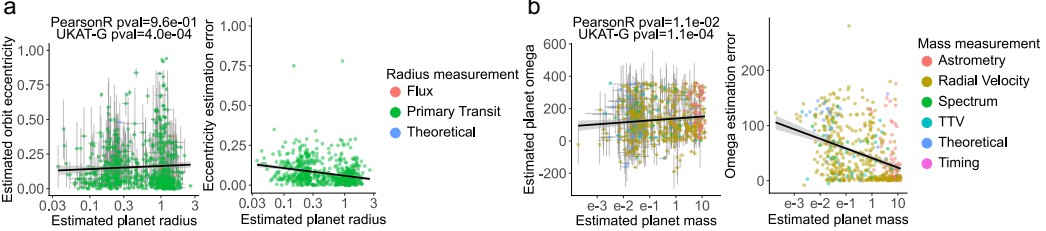

Figure 6: **UKAT reveals associations between exoplanet properties and measurement precision.** (a) Planet radius and orbital eccentricity show no correlation, but smaller planets exhibit higher eccentricity estimation error. (b) Planet mass shows mild correlation with argument of periastron omega, while the association between omega estimation uncertainty and mass is much stronger.

---

[1]Dataset from Kaggle at akashbommidi/exoplanets-dataset/data

## 5 DISCUSSION

As scientific instrumentation increasingly quantifies measurement uncertainty, statistical methods should evolve to effectively utilize this rich information. `UKAT` provides a practical framework for uncertainty-aware statistical inference by treating individual distributions rather than individual data points as the fundamental units of analysis.

The technical foundation of `UKAT`, built upon kernel association testing with Wasserstein distances and energy kernels, demonstrates robust power improvements, particularly under noisy uncertainty estimates. Beyond statistical power gains, `UKAT` can reveal associations between variables and measurement precision that remain invisible to traditional methods. In our LLM example, uncertainty-aware testing uncovered behavioral differences between prompting strategies that accuracy-focused analyses completely missed, illustrating the value of incorporating uncertainty information.

However, `UKAT` faces several limitations. First, performance depends on uncertainty estimation quality, particularly regarding scaling errors (Figure 8). While simulations show robustness to noisy uncertainties, especially for the rank-based `UKAT-R`, poorly calibrated estimates could still mislead inference. Future work may extend upon domain-specific uncertainty calibration and outlier detection approaches. Second, like other kernel association tests, `UKAT`'s computational complexity scales quadratically with sample size due to kernel matrix computations, potentially limiting applications to very large datasets. Advances in scalable kernel methods and low-rank approximations could address this limitation in the future. Third, the asymptotic $\chi^2$ mixture null distribution may be overly conservative for smaller sample sizes (Figure 7), suggesting room for optimizing test procedures, distance metrics, kernel selection, and null approximations to further improve power.

Despite these limitations, `UKAT` represents a meaningful step toward incorporating the uncertainty information available in modern datasets, opening new avenues for scientific discovery through uncertainty-aware statistical analysis.

## 6 ETHICS STATEMENT

This work adheres to the ICLR Code of Ethics. No human subjects or animal experimentation was involved in this study. All datasets used were publicly available and sourced in compliance with their respective usage guidelines and licenses.

## 7 REPRODUCIBILITY STATEMENT

To ensure reproducibility, we provide comprehensive implementation details and experimental protocols. Method implementation is described in detail in Section 3, including all kernel constructions, distance metrics, and statistical testing procedures. Simulation parameters are fully specified in Section 4, with complete data generation processes for both mean and variance testing scenarios. All hyperparameters, sample sizes, and experimental settings are explicitly stated throughout the results section. Code for implementing `UKAT` variants and reproducing all experiments will be made available upon publication.

## 8 LLM USAGE

Large Language Models were used in several capacities during this research: **Writing assistance:** LLMs aided in manuscript preparation, including grammar checking, improving readability and clarity, and enhancing the fluency of various sections. **Experimental subject:** LLMs served as experimental subjects in the confidence analysis study (Figure 5), where we evaluated the effects of different prompting strategies on model self-reported confidence scores. **Literature search:** LLMs assisted with initial literature searches. The Related Work section reflects the authors' own analysis and interpretation of the literature. **Coding assistance:** LLMs provided programming support for implementation tasks, with all code thoroughly tested and validated by the authors.

The authors take full responsibility for all content, analyses, and conclusions presented in this manuscript. All content was carefully reviewed and edited by the authors to ensure accuracy and adherence to scientific standards.

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

# A APPENDIX

## A.1 SUPPLEMENTARY FIGURES

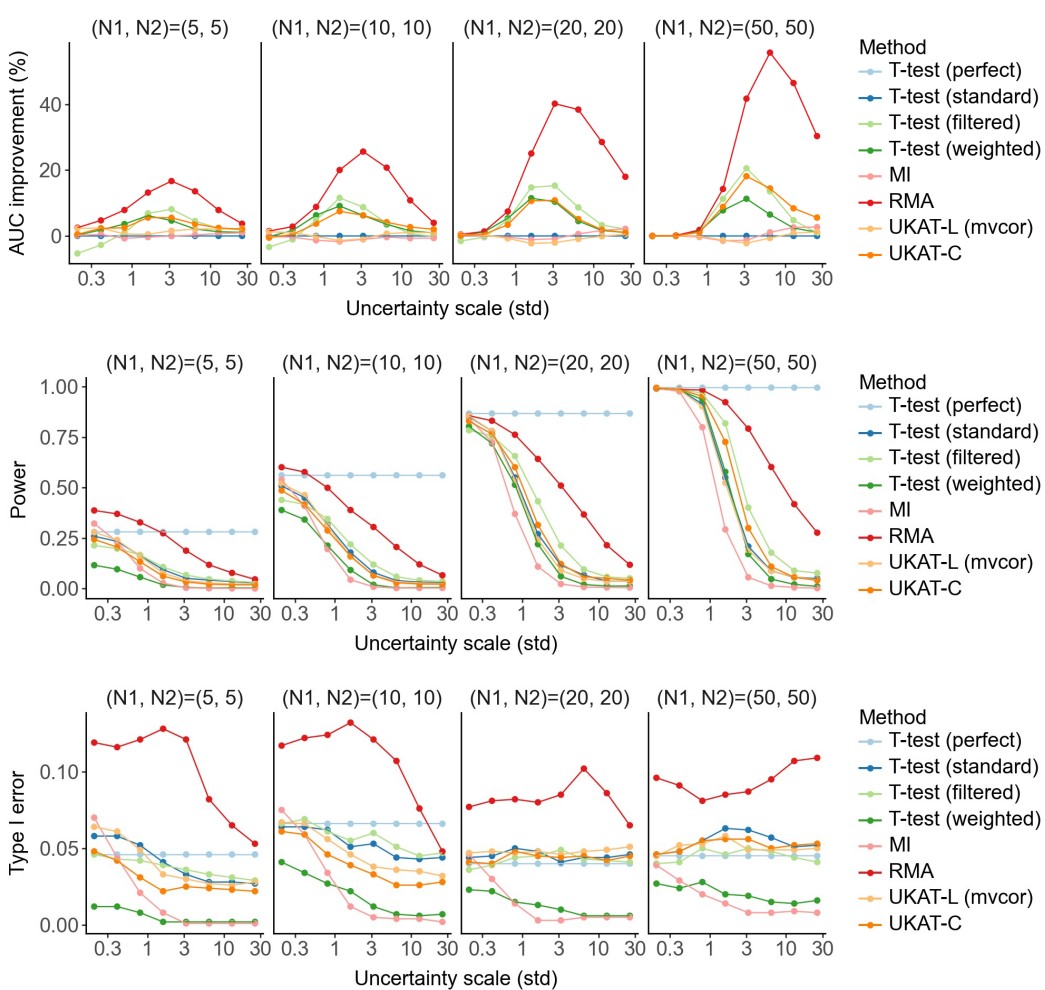

Figure 7: UKAT improvements are robust across sample sizes, related to Figure3.

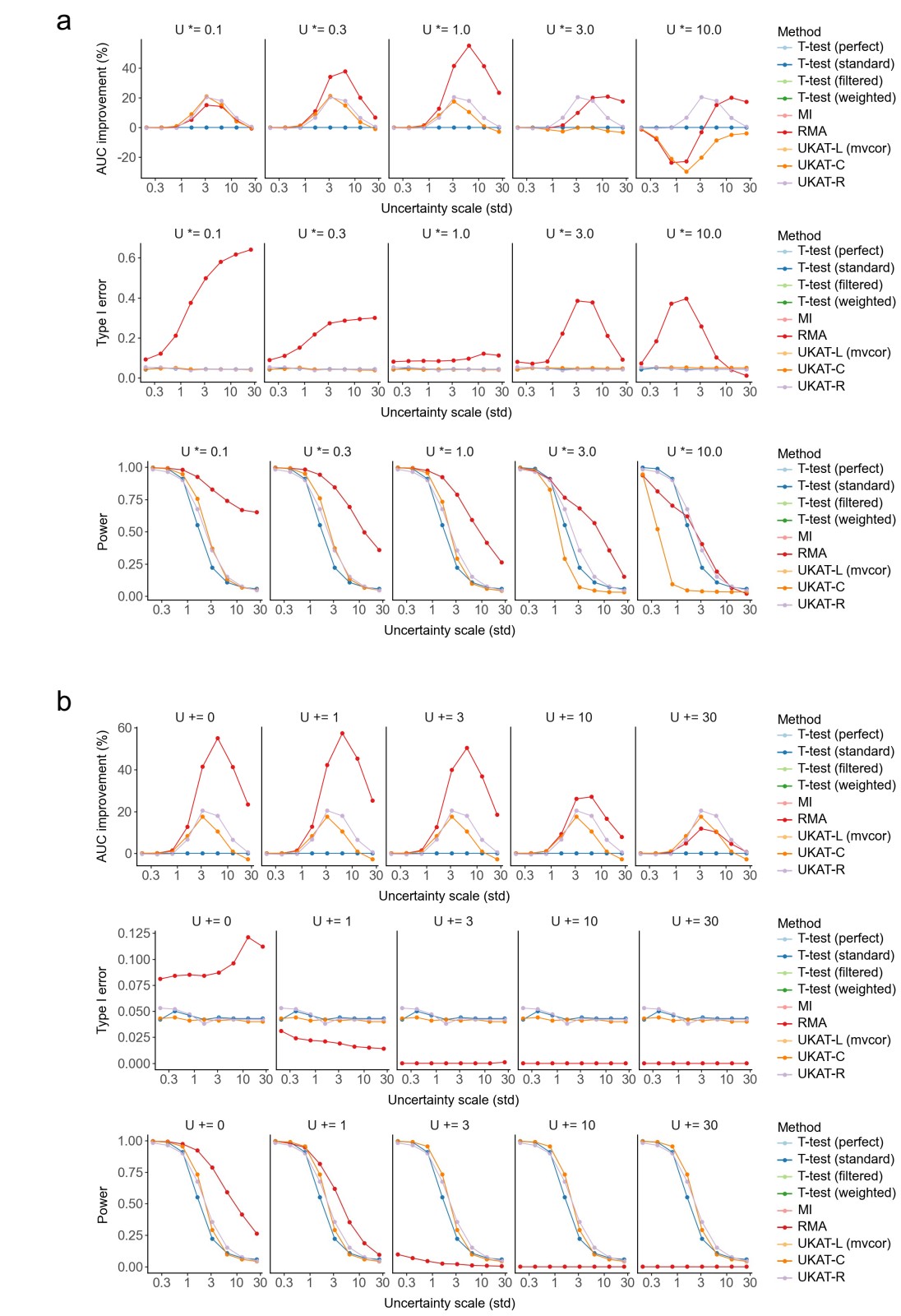

Figure 8: UKAT-C improvements are robust to uncertainty estimation additive biases but not scales, related to Figure3.

