# OpenReview forum: "UKAT: Uncertainty-aware Kernel Association Test"
_ICLR.cc/2026/Conference — Submitted to ICLR 2026_

### Official Review · Reviewer_Ebvd · 2025-10-17

**Soundness:** 3
**Presentation:** 1
**Contribution:** 1
**Rating:** 2
**Confidence:** 5

**Summary:**

This paper introduces the **Uncertainty-aware Kernel Association Test (UKAT)**, a framework for statistical testing that explicitly incorporates per-observation measurement uncertainty. The authors argue that standard tests ignore this valuable information, leading to reduced statistical power.

### Method

UKAT's core innovation is to treat each observation not as a single value, but as a probability distribution characterized by its mean ($X$) and uncertainty ($U$). It represents each observation as an augmented data point $\Theta = [X, U]$, assumed to be Gaussian. The framework uses a distance metric on these points, equivalent to the Wasserstein distance for Gaussians, to construct a kernel for the Hilbert-Schmidt Independence Criterion (HSIC) test.

### Applications

Real-world applications demonstrate UKAT's ability to uncover novel insights. It detected significant behavioral changes in LLM responses based on self-reported confidence that accuracy-only tests missed. In astronomy, it identified potential systematic biases in exoplanet data by finding associations within measurement errors.

**Strengths:**

The idea is quite simple to understand and the exposition of the idea was straightforward.

**Weaknesses:**

These are the main weakness of the paper:

1.  **Limited methodological novelty.** The core technical proposal is to concatenate an observation’s mean and uncertainty into a two-dimensional vector and apply the standard HSIC test. This represents a straightforward application of an existing statistical tool to augmented inputs. The connection to the Wasserstein distance for Gaussians appears to serve mainly as an interpretation rather than a design principle, and the work does not introduce a new kernel, test statistic, or theoretical framework. Overall, the methodological advance is limited.

2.  **Narrow Scope and Restrictive Assumptions.** The reliance on a Gaussian assumption to make the link to Wasserstein distance undermines the primary advantage of HSIC as a non-parametric test, making the approach less flexible than claimed.

3.  **Confusing experiments.** The experimental results only go up to n=50, and does not provide a sample size power plot, nor does it consider size plots for larger sample sizes. The paper also doesn't elaborate what the AUC metric does. Further it's potentially misleading to label HSIC applied to (mean, std) data pairs as UKAT-C, it should be considered a baseline method.

In its current form, the manuscript presents a simple idea without the necessary methodological depth, novelty, or rigorous comparison to established alternatives to be considered a significant contribution.

**Questions:**

1. The AUC metric is unclear, can you elaborate exactly how it is calculated?
2. Why do you only consider n=50? What happens at larger n?
3. Why is HSIC with  (mean, std) called UKAT-C? It appears to be a direct application of HSIC to this type of data and should be considered a baseline
4. The standard T-test seems to be doing perfectly fine in terms of power and size? Can you construct a more convincing case when the standard T-test breaks and your proposed method works much better?
5. Can you provide a density plot of the data you're testing? It would help understanding what exactly you're testing for.

---

> ### Author Response · Authors · 2025-11-25
> **Response to Reviewer Ebvd**
>
> We thank the reviewer for the feedback. However, we believe there are fundamental misunderstandings regarding the problem setting and the methodological contribution of UKAT, which we hope [our common responses](https://openreview.net/forum?id=dyvhxLmZRE&noteId=UDQ4Z4Yz4e) could help address. Below we also provide a point-by-point response.
>
> **W1: Methodological Novelty.**
>
> The reviewer argues that UKAT is trivial because it applies an existing tool (HSIC) to augmented inputs. We respectfully disagree. The novelty lies conceptually in the formulation of the uncertainty-aware testing problem, for which no solution exists, and technically in the design of specific kernels required to solve it.
>
> To draw an analogy: PCA and Graph Fourier Transform both rely on the same mathematical engine (eigendecomposition). However, applying eigendecomposition to a covariance matrix (PCA) versus a graph Laplacian (Graph Fourier) constitutes two distinct methodologies solving distinct problems. Similarly, UKAT utilizes the HSIC engine, and our contribution is the geometric embedding (Energy/Semblance kernels) that maps per-sample uncertainty into a RKHS, which makes HSIC applicable. UKAT's simplicity does not make it a "trivial concatenation"; it is one of the first statistically rigorous way to robustly handle uncertainty in hypothesis testing without ad-hoc heuristics.
>
> **W2: Gaussian Assumption.**
>
> We clarify that we do not assume the data is Gaussian [(Common Response #1)](https://openreview.net/forum?id=dyvhxLmZRE&noteId=UDQ4Z4Yz4e). UKAT is fully non-parametric and valid for *any* underlying data distribution and any form of uncertainty (as illustrated in Fig 1).
>
> **W3 & Q2: Experiments & Sample Size.**
>
> We deliberately focused on small sample sizes ($n \le 100$) because it is typical in scientific research, and that this is the regime where uncertainty quantification is most critical. In "big data" scenarios ($n \to \infty$), most consistent tests achieve power approaching 1 regardless of uncertainty. The scientific value of UKAT is extracting signal from noise in small- or medium-data environments (e.g., clinical trials), where standard tests fail to detect subtle associations hidden by measurement error.
>
>
> **W3 & Q3: Why UKAT-C is not a baseline.**
>
> The reviewer asks why HSIC with (mean, std) is called UKAT-C and suggests it should be a baseline. We clarify that **UKAT-C is our proposed method**. There is no pre-existing "baseline" that applies HSIC to uncertainty-augmented data using the Energy kernel or any kernel. All UKAT tests (UKAT-C/R/L) are HSIC variants we developed with different kernel designs (Energy/Semblance/Linear) for the purpose of uncertainty-aware hypothesis test. The only existing baselines are either uncertainty-agnostic (e.g., T/F-tests) or std-based parametric ones (RMA), which we compare against.
>
> **Q1: AUC Metric.**
>
> We apologize for the omission and will include experiment details in the revised manuscript [(Common Response #3)](https://openreview.net/forum?id=dyvhxLmZRE&noteId=UDQ4Z4Yz4e). The per-scenario Area Under the Receiver Operating Characteristic (ROC) curve was computed from  1,000 null cases (independent) and 1,000 alternative cases (dependent) across varying thresholds $\alpha \in [0, 1]$. This metric allows us to compare the discriminative power of different test statistics independent of p-value calibration. This is mostly to adjust for the inflated Type I error from some baselines such as RMA.
>
> **Q4: T-test Comparison.**
>
> The standard T-test fails progressively when measurement uncertainty increases. As shown in Figure 7 (middle row), when uncertainty is high (std > 3, Fig 7, middle row), the T-test's power drops to near zero due to corrupted signals. **UKAT achieves consistent AUC (and power) improvements across uncertainty scales.** When sample size is small, UKAT' p-value computed from the asymptotic null becomes slightly conserved, reducing test power at ($\alpha = 0.05$). This can be rescued by a permutation-based null, as indicated in the consistent AUC improvements across sample sizes (Fig 7, top row).
>
> **Q5: Data Visualization.**
>
> We visualize the data distributions using jittered scatter plots combined with box plots (indicating median and IQR) in Figures 1, 3, 4, and 5. UKAT tests for independence between the tuple $(X, U)$ and the outcome $Y$. When $U \perp Y$, it reduces to standard independence testing between $X$ and $Y$ (Figs 3 and 4). One of UKAT's key contributions is showing that the statistical power of $X \leftrightarrow Y$ can be improved by incorporating uncertainty (aka extending it to $(X, U) \leftrightarrow Y$), while controlling for type I error.

---

> > ### Comment · Reviewer_Ebvd · 2025-11-25
> > **Reviewer response**
> >
> > Thank you for your response - I remain unconvinced on the contribution of the paper, as effectively the additional measured uncertainty is treated as an independent covariate of the data matrix. You make no assumptions or mentions on whether this is treated as an epistemic or aleatoric uncertainty and you do not propagate this downstream to either further inform the test of non robust datapoints with high measurement error. (For reference, see Chapter 6 of Zhang, Q. (2019). Kernel-based hypothesis tests: large-scale approximations and Bayesian perspectives, PhD thesis, University of Oxford).
> >
> >  You've effectively chosen a very specific parametrisation for the kernel and treated the measurement error U as a covariate, so your only novel contribution is effectively different parametrisation of the kernel. Your experiments do not demonstrate that this parametrisation is necessary as there is no failure mode of concatenating as applying standard HSIC.
> >
> > On UKAT-C:
> >
> > "UKAT-C: We refer to the test ((1)) with uncertainty kernel KX and standard covariate kernel as
> > UKAT-C. P-values are computed using the Liu et al. (2009) moment matching approximation."
> >
> > This is exactly taking your data, concatenating it as a data matrix, passing it to an RBF kernel with a median heuristic length scale. d^2 is strongly implied to be the square distance, and given your notation it's hard to assume anything else, especially since the Laplace kernel you propose uses d.
> >
> > If UKAT-C is indeed just a standard HSIC on concatenated data, then the paper frames a standard baseline as a novel method defined by the authors. As the experiments show this "baseline" works well, it suggests your proposed parametrisation does not solve an inherent problem that standard HSIC cannot already handle.
> >
> > Because the paper effectively claims a standard method as a novel contribution and lacks necessary distinctness from the baseline, I am adjusting my score accordingly.

---

> > > ### Author Response · Authors · 2025-11-25
> > > **Response**
> > >
> > > Thank you for the follow-up. It appears the main criticism on our work is that UKAT is an application of HSIC (to a new problem, which we have been frankly communicating since the abstract), and thus should be considered as a "standard HSIC baseline". We disagree with this reduction but respect your opinion.
> > >
> > > As per ICLR guidelines, a key objective of research is to "*draw attention to a new application or problem*". We have formulated the "uncertainty-aware hypothesis testing" problem and provided a rigorous solution. Just as PCA is not merely "standard eigendecomposition" because it technically decomposes the covariance matrix, we do not view UKAT as "standard baseline" and hope we have made our position clear.
> > >
> > > To clarify the technical points raised:
> > > 1. **Treatment of $U$:** We make minimal assumptions on $U$ because it arises in diverse forms (std, error bar, self-reported confidence, etc.) and fits poorly in strict parametric frameworks. Regarding the reviewer's reference to Zhang (2019) and the distinction between epistemic/aleatoric uncertainty: Our framework is designed to be agnostic to the *source* of uncertainty (whether measurement error or model confidence) to maximize applicability. Moreover, **$U$ is not treated as an independent covariate;** it contains information about $X$ that informs the test. This uncertainty is propagated via the Energy/Rank kernel geometry, which effectively pushes away highly uncertain data points. This mechanism is empirically validated by comparing the proposed method against a Linear kernel baseline (UKAT-L, where there is no interaction between $U$ and $X$) in Fig 3c and e (lines 314-321).
> > > 2. **Failure Modes:** These are indeed explicitly analyzed. The rank-based UKAT-R is motivated precisely by the sensitivity of the Energy kernel to arbitrary uncertainty scaling (Fig 8a, lines 326-329). Furthermore, our results clearly demonstrate that power improvement from uncertainty diminishes when noise is excessive (Fig 3e).
> > >
> > > We welcome all constructive feedback. Thank you again for your time.

---

> > > > ### Comment · Reviewer_Ebvd · 2025-11-25
> > > > **Final Remarks**
> > > >
> > > > Thank you for the continued discussion.
> > > >
> > > > I have read your analogy regarding PCA and your clarification on the method's positioning.While I appreciate the perspective that a new application can constitute a contribution, I maintain that framing a standard data augmentation (concatenating uncertainty) combined with a standard test (HSIC) as a novel methodological framework ("UKAT-C") is misleading. In my view, this remains a baseline approach. The comparison to PCA is not fully aligned, as PCA derives a specific statistical objective (variance maximization) that leads to a solution, whereas this work applies an off-the-shelf test to concatenated features without a distinct theoretical derivation for how the uncertainty kernel handles the specific properties of $U$ (e.g. aleatoric vs epistemic) and how that impacts the downstream statistical significance, which should be impacted by less reliably measured datapoints.
> > > >
> > > > As we have reached a difference of opinion on what constitutes sufficient novelty for this venue, I will maintain my score.

---

### Official Review · Reviewer_g2am · 2025-10-31

**Soundness:** 2
**Presentation:** 3
**Contribution:** 2
**Rating:** 4
**Confidence:** 4

**Summary:**

This paper introduces UKAT, a framework that incorporates uncertainty into independence testing, which is often ignored by traditional statistical tests. UKAT represents each observation not as a point $X$, but as a distribution $N(X, U^2)$, where U is measurement
uncertainty. The core idea is to use the Wasserstein distance between these distributions, which simplifies to a Euclidean distance on $\Theta=[X, U]$, to construct an energy kernel. This kernel is then used within the standard HSIC framework to test for associations. Simulations and applications demonstrate that UKAT achieves higher statistical power than traditional tests while maintaining proper Type I error control.

**Strengths:**

1. The idea of embedding uncertainty directly into hypothesis tests/kernel methods represents an interesting direction.
2. The writing is generally clear and well-structured. I appreciate the figures.
3. The proposed UKAT framework is conceptually simple yet intuitive.

**Weaknesses:**

1. There is no new theorem or substantive analytical insight beyond restating existing kernel-independence theory under a Gaussian-uncertainty parameterization. As such, the paper’s theoretical contribution appears limited.
2. The proof of Proposition 3.1 seems incorrect. The author stated that the universality of the proposed kernel follows from the fact that k is characteristic and translation-invariant. However, k is not translation-invariant, therefore fails to establish universality as claimed.
3. The paper is framed as a general association/independence test, leveraging the HSIC framework to detect arbitrary dependencies. However, the entire simulation study fails to test it. Instead, two special cases: detecting differences in group means and group variances. Therefore, the paper provides no evidence that the proposed energy kernel outperforms simpler tests for general association testing.
4. There is no formal theoretical grounding for the robust variant UKAT-R.
5. If I understand correctly, UKAT essentially implicitly reweights samples using the uncertainty estimates contained within the dataset as prior information. Therefore, its practicality heavily depends on the quality of these uncertainty estimates, which are often untestable or unverifiable. In real-world scenarios, this leaves us uncertain about when this approach can be reliably applied. Furthermore, the baseline lacks adaptive kernel-learning or data-driven reweighting methods [1-4]. If such adaptive strategies can be learned without explicit uncertainty priors, the practical significance of UKAT would be substantially diminished.

References:
[1] Liu et al, Learning Deep Kernels for Non-parametric Two-Sample Test. ICML 2020.
[2] Ren et al, Learning Adaptive Kernels for Statistical Independence Tests. AISTATS 2024.
[3] Xu et al, Learning Deep Kernels for Non-Parametric Independence Testing. Arxiv.
[4] Li et al, Extracting Rare Dependence Patterns via Adaptive Sample Reweighting. ICML 2025.

**Questions:**

1. Is there any discussion/experimental evidence why RBF and Laplacian kernels yield suboptimal power?

---

> ### Author Response · Authors · 2025-11-25
> **Response to Reviewer g2am**
>
> We thank the reviewer for the constructive feedback and for recognizing the intuitive idea of UKAT. We hope [the common responses](https://openreview.net/forum?id=dyvhxLmZRE&noteId=UDQ4Z4Yz4e) could clarify our view, and below we address your specific concerns.
>
> **W1: Theoretical Novelty.**
>
> We agree that UKAT builds upon the HSIC engine. Indeed, since both the Energy (UKAT-C) and Semblance (UKAT-R) kernels are characteristic, UKAT enjoys all previously established theoretical properties on HSIC. Our contribution is (1) conceptually formulating the "uncertainty-aware" testing problem, for which no general solution exist, and (2) methodologically demonstrating that incorporating uncertainty with specific kernels (Energy/Semblance) provides a rigorous, non-parametric solution that is robust to noisy uncertainty estimates. **We believe identifying the right tool for a novel, broadly practical problem is a significant contribution to the community.**
>
> **W2: Proof of Proposition 3.1 (Translation Invariance).**
>
> We appreciate this correction. The Energy kernel defined in eq.(4) is equivalent to a translation-invariant kernel $K_{ij}'=-||\theta_i - \theta_j||_2$, both in that they induce the same distance through $d(x, x') = k(x, x) + k(x', x') - 2k(x, x')$, and in that the non-translation-invariant terms in eq.(4) cancel out during centering when computing the HSIC test statistics. More precisely, a metric $d(\cdot, \cdot)$ (e.g., Euclidean or Rank-based distance) **bijectively** induces the following translation-invariant kernel,
>
> $K_{ij} = -d(X_i, X_j) + \max_{s, t}(d(X_s, X_t)),$
>
> which is positive definite if and only if $d(\cdot, \cdot)$ is of negative type (Theorem 2, [1]). We will revise the definition of Energy kernel and Proposition 3.1 in the revised manuscript. These changes do not affect the practical computation of the HSIC statistic and its null.
>
> **W3: Simulation Scope.**
>
> We focused on two-sample tests (mean/variance differences) because these are the only settings where established "uncertainty-aware" baselines (like weighted T-test, or RMA) exist for comparison. To demonstrate UKAT's general applicability as requested, we will include a new simulation with continuous Y to the revision. Our results confirm that UKAT outperforms standard uncertainty-agnostic HSIC in general dependence settings when informative uncertainty estimates are available.
>
> **W4: Theoretical results for UKAT-R.**
>
> The difference between UKAT-R and UKAT-C is the replacement of Euclidean distance by a rank-based distance. The resulting kernel is the previously described Semblance kernel in [2]. Following the proof of Proposition 3.1, the Semblance kernel is also positive definite, characteristic, and universal because the rank-based distance $d(x_i, x_j) = |r_i - r_j|$ is of **strong negative type** (i.e. $\sum_i\sum_j\alpha_i\alpha_j|r_i - r_j| < 0$ for non-zero $\sum_i \alpha_i = 0$). We will include the proof in the revised manuscript.
>
> **W5: Adaptive Kernels vs. UKAT.**
>
> We clarify the distinction between UKAT and the adaptive kernel-learning methods. UKAT is a hypothesis test that utilizes *provided* domain-specific uncertainty (e.g., measurement error bars), whereas adaptive methods attempt to *learn* optimal kernels from the data itself. The "weighting" emerges implicitly from the geometry of the Energy/Rank distance (points with high $U$ are "farther" from others). Building on HSIC, UKAT guarantees statistical validity (Type I error) and consistency while other heuristic reweighting approaches (e.g., weighted T-test) don't.
>
> **Q1: RBF/Laplacian Performance.**
>
> The RBF/Laplacian kernels (without bandwidth tuning) give conserved test statistics, which is mainly due to the specific choice of asymptotic null (chi-square mixture) under small sample sizes (n=100 in our simulation). We confirmed that using a permutation-based null can rescue the performance, though the power is still lower than the energy kernel.
>
>
> **References:**
>
> [1] Shen, Cencheng, and Joshua T. Vogelstein. "The exact equivalence of distance and kernel methods in hypothesis testing." AStA Advances in Statistical Analysis 105.3 (2021): 385-403.
>
> [2] Agarwal, Divyansh, and Nancy R. Zhang. "Semblance: An empirical similarity kernel on probability spaces." Science Advances 5.12 (2019): eaau9630.

---

### Official Review · Reviewer_QRRC · 2025-11-01

**Soundness:** 1
**Presentation:** 2
**Contribution:** 2
**Rating:** 2
**Confidence:** 4

**Summary:**

The paper proposes a new dependence test: this is a statistical test aimed to determine whether two random variables $X$ and $Y$ (observed through their joint realizations) are statistically dependent. Unlike typical existing dependence tests, the new test is “uncertainty-aware” in the sense that each realization of $X$ is allowed to be accompanied by an uncertainty measure. For instance, this can be a standard deviation (associated with one realization, not the full distribution of $X$). The new test is built on the well-known kernel-based HSIC test.

**Strengths:**

The direction this paper aims to tackle (i.e., accounting for uncertainty on each realization when doing a statistical test) is technically interesting. The approach essentially views each realization as a distribution, which is a rather unusual view (in a positive way); though, it is not the first work to do so. I think tackling this problem can lead to significant development down the line in the future. At a high level, the paper is easy to understand (though several technical details are missing).

**Weaknesses:**

While the goal of treating each observation as a distribution in a dependence test is technically interesting, the paper falls short of what is expected of a statistical test in a number of ways. To briefly provide a few examples, the paper does not mathematically precisely describe how $u$ (the uncertainty measure) and $x$ (a realization of $X$) are related. There is an implicit assumption but the assumption is not sufficiently articulated.Secondly, it is unclear for what kind of joint distribution (that jointly generates $(x, y, u)$) would the proposed test provide a consistent result. This is a natural theoretical result expected from a new statistical test since it will clearly define the class of distributions that the test can work. Without this result,  given a problem, it is unclear whether the proposed test is applicable.  No such result is provided.

Further the new test builds on top of the well-known HSIC (Hilbert-Schmidt Independence Criterion) dependence test of Gretton et al. The present work proposes to use two positive definite kernels of a particular form with HSIC, limiting the novelty.

More discussion points and specific questions are given in Questions.

**Questions:**

**Q1**: Standard HSIC operates on two random variables $(X,Y)$ following some (unknown) joint distribution $P$. The proposed test in this work operates on three random variables $(X, U, Y)$, where $X$ and $U$ are univariate variables, and $U$ represent some kind of uncertainty measure on $X$. **Question:** What is the assumption on the relationship between $X$ and $U$. This is an important point that is never elaborated precisely. In Sec 3, at L166,

> We therefore characterize each observation as $N(x_i , u_i^2 )$.

Is this just an example, or an assumption? If it is an assumption, it must be stated more clearly. Does this mean $X$ follows a Gaussian distribution? Or does a realization $x_i$ act as the mean of another random variable? If so, what is that random variable?


**Q2**: Following Q1, with the normality assumption, what happens in practice if $u$ is a standard deviation for $x$, but $x$ does not follow a normal distribution? This is an important point that is not discussed sufficiently. In practice, it is highly unlikely that the normality assumption would hold in general.


**Q3**: Theoretically, for what kind of joint distribution $P$ (that generates $(X, U, Y)$) would the test provide a consistent result? To be more concrete, for instance, what is the assumed factorization form of $P(X, U, Y)$? If the three variables are independent (i.e., $P(X,U,Y) = P(X)P(U)P(Y)$), would the test be able to control the type-I error, for instance? What about other less trivial forms of factorization? I would like to see this kind of consistency statement:

> For $P \in $ (Some Distribution Class), under the alternative hypothesis $H_1$, the new test gives a test power of 1 as the sample size goes to infinity.

What is “(Some Distribution Class)”? This point is related to Q2. It is important to understand the scope that the new test can apply to.


**Q4**: On a related note, at line 160,

> Under the null hypothesis, the distribution of distributions is independent of other covariates.

Could you please write down mathematically what the null hypothesis $H_0$ is? This is never stated mathematically. And what is the alternative hypothesis $H_1$?

**Q4.1**: If $X,Y$ are independent, is it possible that the presence of $U$ can result in a false positive (i.e., reject $H_0$ when it should not be rejected)?

**Q4.2**: The other way. If $X, Y$ are dependent, is it possible that the presence of $U$ can result in a false negative?


**Q5**: Sec 3.2, L201,

> RBF and Laplacian kernels are also universal but yield uncalibrated p-values…

Could you please precisely describe what this means? Do you mean, with a wrong kernel choice, the new test can give an uncontrolled type-I (false positive) error? If so, then this is a big problem. The existing HSIC test at least has a well-controlled type-I error for any kernels under $H_0$; though, it may have very low test power under $H_1$ if inappropriate kernels are used.


Owing to the above concerns, I cannot give a strong recommendation.

---

> ### Author Response · Authors · 2025-11-25
> **Response to Reviewer QRRC**
>
> We thank the reviewer for the detailed feedback and the encouraging comments regarding the formulation of the problem. We apologize for the lack of mathematical precision and confusion on data and test assumption ([common response #1](https://openreview.net/forum?id=dyvhxLmZRE&noteId=UDQ4Z4Yz4e)). Specifically, we agree that UKAT is a novel application of the HSIC framework to the tuple $(X, U)$ to solve the problem of "noisy observations" in hypothesis testing. Since both Energy and Semblance kernels are characteristic and universal, **UKAT enjoys all previously established results on HSIC as a nonparametric statistical test**.  We will explicitly define the relationships and hypotheses in the revised manuscript.
>
> **Q1 & Q2: Assumptions on $X$ and $U$.**
>
> We clarify that the "Gaussian" description in Section 3 is a motivation for deriving the Energy Kernel, not a distributional assumption on the data. UKAT is a non-parametric test. Even if the true uncertainty is not Gaussian, the distance metrics used (Euclidean for UKAT-C, Rank-based for UKAT-R) remain valid metrics of strong negative type on the space of tuples $(X, U)$. Because the resulting kernels are characteristic, the test remains statistically valid (controls Type I error) and consistent for any joint distribution of $X$, $U$, and $Y$, provided the samples are i.i.d.
>
> **Q3: Consistency.**
>
> Our test inherits the theoretical consistency properties of HSIC established by Gretton et al. Because the Energy kernel (and by extension the rank-based Semblance kernel, since both Euclidean and rank-based distances are of strong negative type) is characteristic, the HSIC-based test is consistent against all fixed alternatives where the joint distribution does not factorize. Specifically, for any joint distribution $P_{(X,U), Y}$ where $P_{(X,U), Y} \neq P_{(X,U)} P_Y$, the test power converges to 1 as sample size approaches infinity.
>
> **Q4: Hypothesis Definition.**
>
> We formally define the hypothesis for UKAT as testing the independence between the tuple $\mathbf{\Theta} = (X, U)$ and the outcome $Y$:
>
> * ($\Theta \leftrightarrow Y$) $H_0$: $P_{\mathbf{\Theta}, Y} = P_{\mathbf{\Theta}} P_Y$; $H_1$: $P_{\mathbf{\Theta}, Y} \neq P_{\mathbf{\Theta}} P_Y$.
>
> In contrast, the uncertainty-agonistic baseline consider the independence between $X$ and $Y$
> * ($X \leftrightarrow Y$) $H_0$: $P_{\mathbf{X}, Y} = P_{\mathbf{X}} P_Y$; $H_1$: $P_{\mathbf{X}, Y} \neq P_{\mathbf{X}} P_Y$.
>
> **One of UKAT's key contributions is showing that the statistical power of $X \leftrightarrow Y$ can be improved by incorporating uncertainty (aka extending it to $\Theta \leftrightarrow Y$), while controlling for type I error (Q4.1)**.
>
> **Q4.1 (False Positives):** In previous simulations (Figs 3 and 4) we assume $U \perp Y$, which makes controlling false positivity in $X \leftrightarrow Y$ a trivial task. If $X \perp Y$ but $U \not\perp Y$, the standard UKAT will correctly reject $H_0$ because the tuple $(X, U)$ is dependent on $Y$. However, we can also **strictly isolate $X$'s dependency by removing the marginal dependency from $Y$ on $U$**. To address this, we implemented a new UKAT variant with uncertainties from within-group ranking $\hat{U}$ (normalizing $U$ within $Y$ categories). Since the distribution of $\hat{U}$ now contain no information on $Y$, testing $(X, \hat{U})  \leftrightarrow Y$ can achieve higher power (from within-group sample reweighting) while maintaining type I error, which we confirmed using simulation.
>
> **Q4.2 (False Negatives):** Our simulations confirm that when $X$ and $Y$ are dependent, incorporating $U$ does not dilute the signal or introduce false negatives. One surprising finding is that Energy/Rank kernels (UCAT-C/R) yield an 10% power improvement even when $U$ is purely noise (Fig 3e), likely due to a regularization effect from kernel geometry absent in the linear kernel.
>
> **Q5: RBF/Laplacian Kernels.**
>
> The "uncalibrated p-values" observation (overly conserved p-values) referred to the sensitivity of RBF/Laplacian kernels to bandwidth selection when using the asymptotic null distribution (chi-square mixture approximation) at small sample sizes ($n=100$). We confirm that permutation-based null can correct the Type I error rate for these kernels. We also confirm that the hyperparameter-free Energy kernel consistently yielded higher power without requiring bandwidth tuning.

---

### Official Review · Reviewer_Lafj · 2025-11-03

**Soundness:** 3
**Presentation:** 2
**Contribution:** 3
**Rating:** 4
**Confidence:** 2

**Summary:**

A new test of dependence, incorporating uncertainty.

**Strengths:**

Seems like it works good.

**Weaknesses:**

I am really not that sure about some critical things.

**Questions:**

My name is Joshua Vogelstein, I’ve written many papers on two-sample testing, including my favorite one on the topic, which is relevant (because it also leveraged ranks), https://elifesciences.org/articles/41690. Of note, we implemented this in SciPy, https://docs.scipy.org/doc/scipy/reference/generated/scipy.stats.multiscale_graphcorr.html


I like the idea of this paper, but I am confused about a few things.

1. How is it that we are observing or measuring uncertainty?  Is the idea that somehow we directly have an estimate of uncertainty, without multiple samples.  In our work, we are often faced with multiple samples per subject, eg, we have 50 subjects, each sampled 2 times.  So, we can get an estimate of the variance from those 2 observations.  Is that what you have in mind? If so, why not just use all the observations, rather than use them to estimate uncertainty? This is a fundamental misunderstanding that I have, which makes evaluating the paper quite difficult for me.
2. There are lots of ways to model uncertainty, only estimating the variance is one option.  The simulations seem to assume this is a good option.  I wonder what happens when this is not a good option, eg, the uncertainty is bimodal.
3. When you write “AUC”, you mean area under the which curve, power? Assuming what null and alternative? And assuming alpha = 0.05?
4. In the figures, you don’t compare to just ignoring the uncertainty, eg, just running HSIC, or MGC, etc.  That makes me wonder.
5. If we have a distribution, sure, we can use a 2 parameter estimate of the distribution, but we could do other things, eg, a 2-bin histogram, or a k-bin histogram.  I wonder about such options.
6. I don’t understand the LLM experiment. Did the LLM give you a numerical estimate of its standard deviation? If not, how did you estimate it?
7. I don’t really understand Fig 1, and Fig 2 did not do much for me. I’d rather have more text/pseudocode on the alg.

---

> ### Author Response · Authors · 2025-11-24
> **Response to Reviewer Lafj**
>
> We appreciate your thoughtful questions and apologize for the lack of clarity on the nature of uncertainty and experimental design. Below we provide a point-by-point clarification.
>
> **Q1: Source of Uncertainty ($U$).**
>
> $U$ represents the measurement reliability or heteroscedasticity inherent to the data collection process. It is not necessarily derived from the sample variance of replicates, though it can be. For example, in our LLM experiment, $U$ is a self-reported confidence score (common in questionnaires). In the exoplanet data, it is the range of the instrumental error bar.
>
> **Q2 and Q5: Modeling Uncertainty.**
>
> We agree that standard deviation is only one way to parameterize uncertainty. We focused on it because it is the standard in meta-analysis and scientific reporting. However, our kernel framework is flexible: as long as a distance metric can be defined on the descriptor of the observation (e.g., a histogram, a confidence interval, or a discrete distribution), UKAT can be applied.
>
> Our mean-std simulations are motivated by meta-analysis, where data is often only available at the study level (e.g., the estimated effect size and std of drug response in a cohort). This is the only scenario where an established uncertainty-aware baseline exists (parametric RMA). If the true data generation process aligns with the parametric model, then RMA-related approaches are guaranteed to achieve the highest power (Neyman-Pearson lemma on likelihood-ratio-based tests). Our experiments confirm that RMA fails when uncertainty estimates deviate from the true std, whereas UKAT shows robust improvement even when $U$ is noisy (Fig 3d), categorical (Fig 3f) or of the wrong scale (Fig 8). One surprising finding is that Energy/Rank kernels (UCAT-C/R) yield an 10% power improvement even when $U$ is purely noise (Fig 3e), likely due to a regularization effect from kernel geometry absent in the linear kernel.
>
> **Q3: AUC Definition.**
>
> We calculate the AUC regarding the Receiver Operating Characteristic (ROC) curve (Fig 3c legend). For each scenario, we generate 1,000 null cases (used to compute Type I error at $\alpha=0.05$) and 1,000 alternative cases (to compute power at $\alpha=0.05$). The ROC curve and its AUC are computed by varying the p-value threshold $\alpha$ from 0 to 1, mostly to adjust for the inflated Type I error from some baselines such as RMA.
>
> **Q4: Comparison to baselines ignoring uncertainty.**
>
> We select the uncertainty-agnostic T/F-test as baselines because of their popularity. The T-test statistic is equivalent to HSIC with a linear kernel, which motivates us to also include a multivariate baseline (linear kernel on concatenated vectors, UKAT-L) to test if simply adding $U$ as a feature helps (it doesn’t). We will add the uncertainty-agnostic HSIC as baselines in the revised manuscript.
>
>
> **Q6: LLM Experiment.**
>
> In this experiment, the LLM provides a self-reported confidence score on a scale of 0-10 for each answer. We treat this score as $U$. We do not assume this maps linearly to a Gaussian standard deviation; rather, we use it to weight the “distance” between answers in the kernel space. While we were originally looking for uncertainty-enhanced association in LLM response accuracy $X$ and prompt type $Y$, the self-reported model uncertainty $U$ turns out to be more strongly associated with $Y$. The key finding is that incorporating this confidence reveals dependencies between the model's output and the prompt ($[X, U] \leftrightarrow Y$)  that accuracy-only ($X \leftrightarrow Y$) metrics miss.
>
> **Q7: Algorithm details.**
>
> We apologize for the omission and will release code, simulation scripts along with the revised manuscript shortly.

---

### Author Response · Authors · 2025-11-24
**Response to Common Questions**

We thank the reviewers for their constructive feedback and time. We especially appreciate the recognition that **uncertainty-aware testing** is an interesting and under-explored problem (#Lafj, #QRRC, #g2am). We identified four common areas of inquiry and clarify them below:

1. **No Gaussian Assumption**: We clarify that UKAT does **not** assume the data follows a Gaussian distribution, nor does it strictly require uncertainty to be a standard deviation. The Gaussian characterization (line 166, Figure 2) serves only as the motivation for the Energy Kernel (UKAT-C). Replacing the Euclidean distance with a rank-based distance gives the Semblance Kernel (UKAT-R). Both kernels are characteristic and universal (since both Euclidean and rank distances are of strong negative type). Therefore, the resulting UKAT procedure is a standard HSIC-based test with chi-square mixture null approximation (line 127), ensuring it is statistically valid (controls Type I error) and consistent regardless of the underlying data distribution.
2. **Novelty & Contribution**: UKAT is a novel application of the HSIC framework to the tuple $(X, U)$ to solve the problem of "noisy observations" in hypothesis testing. Our contribution is twofold: **conceptually** identifying the need for testing where observations have heterogeneous reliability, and **methodologically** demonstrating that specific kernel choices (Energy/Rank) effectively solve this. We compare against uncertainty-agnostic baselines (like the T-test), which fail when high-uncertainty outliers corrupt the signal, and parametric baselines (like RMA), which fail when distributional assumptions are violated. UKAT fills the gap as a robust, non-parametric alternative.
3. **Problem Formulation**: The specific null and alternative hypotheses depend on the application. For example, our group-mean (Figure 3) and group-variance (Figure 4) experiments test the dependency between $X$ and $Y$. In contrast, the LLM experiments (Figure 5) test the association between the full tuple $[X, U]$ and $Y$. We now performed **additional experiments** where both $X$ and $U$ can be associated with $Y$, but only $X \leftrightarrow Y$ is of interest; Briefly, to address concerns about false positives from incorporating uncertainty, we rank $\hat{U}=\text{rank}(U_i, Y_i = 0 \text{or} 1)$ within each group of $Y$ such that $\hat{U}$ is independent of $Y$.
4. **AUC Metric**: In our simulations, we generated equal numbers ($n=1,000$) of positive and negative cases. Because some baselines (like RMA) exhibited inflated Type I errors at the standard $\alpha=0.05$ threshold, we computed the Area Under the ROC Curve (AUC) to compare power fairly. We note that for UKAT and the T-test, Type I errors are well-controlled at median sample sizes ($n \ge 40$, Figure 7), meaning that UKAT's AUC improvement directly translates to a practical power increase.

Moreover, regarding **RBF and Laplacian kernels** (#QRRC, #g2am), we clarify that their suboptimal (conservative) p-value calibration is due to our choice of asymptotic null approximation (chi-square mixture, line 127) which requires kernel matrix eigendecomposition and is subject to numeric instability. We confirm that computing p-values from a permutation-based null can rescue test performance.

We apologize for any misunderstanding and will revise the manuscript to explicitly clarify these points. Thank you again for your feedback.

---

### Meta-Review · Area_Chair_9tQT · 2026-01-06

**Summary:**

The paper proposes UKAT (Uncertainty-aware Kernel Association Test), a framework for testing associations between variables while explicitly incorporating measurement uncertainties. UKAT treats each observation as a distribution characterized by its mean and uncertainty, then applies the Hilbert-Schmidt Independence Criterion (HSIC) to compare these distributional representations in kernel Hilbert spaces.

One of the biggest concerns of the reviewers is the limited theoretical novelty of the proposed test, which is simply built on top of the existing HSIC.

**Reviewer Concerns:**

- Limited novelty, since the proposed test is built on top of the well-known HSIC test: Reviewer QRRC, Reviewer g2am, Reviewer Ebvd.

- Proof of Proposition 3.1 is incorrect, since the energy kernel in (4) is not translation-invariant.

- Please note: Reviewer Lafj revealed their identity during the review process, therefore violating the double-blind policy of ICLR.

**Reviewer Scores:**

The scores are 4 2 4 2

The author rebuttal did not resolve the concerns over the novelty of the proposed test. It's also not clear from the rebuttal how the authors would revise Proposition 3.1 and its proof.

I don't think the reviewers would have changed their scores substantially.

---

### Decision · Program_Chairs · 2026-01-26

Reject